# Towards integrated tunable all-silicon free-electron light sources

Charles Roques-Carmes [1,5], Steven E. Kooi [2,5], Yi Yang [1], Aviram Massuda [1], Phillip D. Keathley [1], Aun Zaidi[1], Yujia Yang[1], John D. Joannopoulos[2,3], Karl K. Berggren[1], Ido Kaminer [3,4] & Marin Soljačić [1,3]

Extracting light from silicon is a longstanding challenge in modern engineering and physics. While silicon has underpinned the past 70 years of electronics advancement, a facile tunable and efficient silicon-based light source remains elusive. Here, we experimentally demonstrate the generation of tunable radiation from a one-dimensional, all-silicon nanograting. Light is generated by the spontaneous emission from the interaction of these nanogratings with low-energy free electrons (2–20 keV) and is recorded in the wavelength range of 800–1600 nm, which includes the silicon transparency window. Tunable free-electron-based light generation from nanoscale silicon gratings with efficiencies approaching those from metallic gratings is demonstrated. We theoretically investigate the feasibility of a scalable, compact, all-silicon tunable light source comprised of a silicon Field Emitter Array integrated with a silicon nanograting that emits at telecommunication wavelengths. Our results reveal the prospects of a CMOS-compatible electrically-pumped silicon light source for possible applications in the mid-infrared and telecommunication wavelengths.

[1] Research Laboratory of Electronics, Massachusetts Institute of Technology, 50 Vassar Street, Cambridge, MA 02139, USA. [2] Institute for Soldier Nanotechnologies, NE47, 500 Technology Square, Cambridge, MA 02139, USA. [3] Department of Physics, Massachusetts Institute of Technology, 77 Massachusetts Avenue, Cambridge, MA 02139, USA. [4] Department of Electrical Engineering, Technion–Israel Institute of Technology, 32000 Haifa, Israel. [5] These authors contributed equally: Charles Roques-Carmes, Steven E. Kooi. Correspondence and requests for materials should be addressed to C.R.-C. (email: chrc@mit.edu)

The lack of an efficient silicon light source[1,2] is often pointed to as the primary reason for the comparatively slow development of silicon photonics in the industry[3–5], in contrast to the rate of progress in compound semiconductor photonics or silicon microelectronics. However, silicon remains the preferred platform for many industries such as optical tele-communications, high-performance optical computing, and sensing[6]. Thus, there is a great need for a tunable, integrated silicon light source at telecommunication wavelengths (around 1550 nm). The realization of a scalable, energy-efficient silicon-based light source could find broad applicability in areas such as optoelectronic very-large scale integration (VLSI), mid-infrared sensing, optoelectronic displays, and lighting. Despite a genuine interest from the community, limited efficiencies have been realized in silicon Light-Emitting Diodes (LEDs)[7,8] with engineered band structures as only partial elimination of non-radiative decay processes has been achieved[7]. The key limitations for such applications are that radiated photons must still abide by the transition rules of silicon's indirect bandgap, and the short life-time of non-radiative processes such as Auger recombination. To surpass the limited tunability and low efficiencies of crystalline silicon's intrinsic radiation, more sophisticated fabrication techniques were developed to extract light from modified forms of bulk silicon[9], such as porous silicon[10] and silicon nanocrystals[11]. However, fabricating these low-dimensional silicon systems is impeded by both fundamental and practical challenges such as high annealing temperatures[10,11] which prevent their integration into existing CMOS fabrication processes.

While directly extracting light from a CMOS-compatible silicon substrate seemed to be an intractable task, the fast development of photonic interconnects[12,13] has enabled the development of all-silicon lasers in the telecommunication spectral window relying on stimulated Raman scattering[14,15] or gain in silicon nanocrystals[16]. Even though these devices were shown to operate with relatively high efficiencies, broad operation bandwidth[17], and low bias thresholds[15], an off-chip external optical pump source was still required. A natural solution to bypass the use of bulk silicon is to leverage the superior optoelectronic properties of III–V semiconductors[18–22]. However, an electrically pumped all-silicon source—which would benefit from the low fabrication costs of silicon devices into VLSI[5]—has yet to be demonstrated.

Free-electron sources[23–25], especially in the context of Smith-Purcell (SP) radiation[26], are natural candidates to address this challenge, thanks to their exceptional tunability. However, Smith-Purcell radiation from dielectric substrates, let alone silicon, has not been utilized so far. In contrast, a related effect, the laser acceleration of particles interacting with confined modes in dielectric structures[27], has been widely studied. More recently, there has been a growing interest in the study of incoherent[28] and coherent[29] (transition radiation) cathodoluminescence in dielectrics and semiconductors, with novel experimental techniques to disentangle their relative importance[30,31]. The lack of work in dielectrics and semiconductors for radiation generation from free electrons incident at a grazing angle[32] may originate from the fact that the SP effect was first observed in metallic gratings and was subsequently explained as the constructive interference from the periodic motion of free currents along the surface[33], or as the motion of image charges in a perfect conductor[26].

It has recently been predicted that dielectrics, and generally low-optical-loss materials could not only emit SP radiation, but also outperform metals by judicious design of phase-matching conditions of the electron excitation with high-$Q$ resonances[34,35]. Coincidentally, the prospects of all-silicon free-electron-driven sources are enhanced by the recent development of high-throughput, low bias voltage, densely integrated silicon-gated

Field Emitter Arrays (FEA)[36–38], whose performance surpasses that of their metallic counterparts (Spindt-type emitters).

In this Article, we experimentally demonstrate the generation of tunable radiation from all-silicon nanogratings over the 800–1600 nm wavelength window. In our proof-of-concept experiment, free electrons with kinetic energies in the range from 2–20 keV pass in close proximity to a nanograting in a modified Scanning Electron Microscope (SEM), thus producing SP radiation. Similar electron energies are achievable with existing on-chip electron sources[36,37,39–41] and are here experimentally utilized to excite all-silicon nanogratings. In addition, we theoretically investigate the feasibility of an all-silicon radiation source, which integrates a silicon on-chip gated FEA with a silicon nanograting. We theoretically predict power efficiencies >10% might be attainable by engineering the electron beam and its coupling to photonic modes. Taken together, our observation and analysis pave the way for an electrically pumped all-silicon source for potential applications in the near-infrared, and the telecommunication wavelengths.

## Results

**Tunable emission from silicon over the 800–1600 nm wavelength range.** A schematic of a free-electron-driven silicon radiation source is depicted in Fig. 1a: an electron source (e.g., the electron gun of an SEM, or potentially an on-chip gated FEA or a hot field emitter) produces an electron beam passing in a close proximity to a silicon nanograting. The interaction of free-electrons with periodic structures induces the emission of tunable radiation. This emission is known as the SP effect and follows the well-known energy-angle relation[26] $\lambda = L(\frac{1}{\beta} - \cos\theta)$ (where $\lambda$ is the radiated wavelength, $L$ the period of the structure, $\beta = v/c$ the normalized velocity of the electron and $\theta$ the observation angle, measured with respect to the direction of electron propagation). By tuning the electron kinetic energy, we record radiation spanning the 800–1600 nm wavelength range—which encompasses the entire telecommunication wavelengths window—with all-silicon nanogratings.

We record spontaneous emission from all-silicon samples with respective periods of 278 nm (Fig. 2a) and 139 nm (Fig. 2b). The output power and the incident electron beam current are experimentally measured. For each sample, the spectral efficiency (normalized by the incident electron beam current) is plotted for various electron kinetic energies in Fig. 2. Our experimental data (top) is compared to time-domain simulation data (bottom). We notice that, at a given wavelength (e.g., the same ratio $L/\beta$), a narrower spectral bandwidth is observed for lower kinetic energies and smaller grating periods[42] (for instance, one can compare the two spectral line shapes plotted in green in Fig. 2, with respective center wavelength ± FWHM/2 of 1137 ± 96/2 nm (16 keV electrons impinging on 278-nm-period grating) and 1090 ± 30/2 nm (4 keV electrons impinging on 139-nm-period grating)). The bandwidth narrowing for slow electrons can be derived from the Smith-Purcell dispersion relation[42] and observed with adequate collection optics. For electrons in the far-field away from the structure, the generated radiation intensity scales as[34,43] $\exp(-\frac{z}{h_{\text{eff}}})$—with $z$ being the electron-beam height above the grating surface. The characteristic decay length of the electron nearfield $h_{\text{eff}} = \lambda\beta\gamma/4\pi$ is also proportional to the period $L$, as evident from the SP formula[26]. Therefore, we expect the output power to be considerably smaller for shorter grating periods in similar experimental conditions[34].

As shown in Fig. 3a, we have modified a SEM[44,45] to record the radiation. Electrons interact with the sample at a grazing angle of ~1°. This configuration allows electrons to pass close to the nanograting so they interact with numerous unit cells. We use

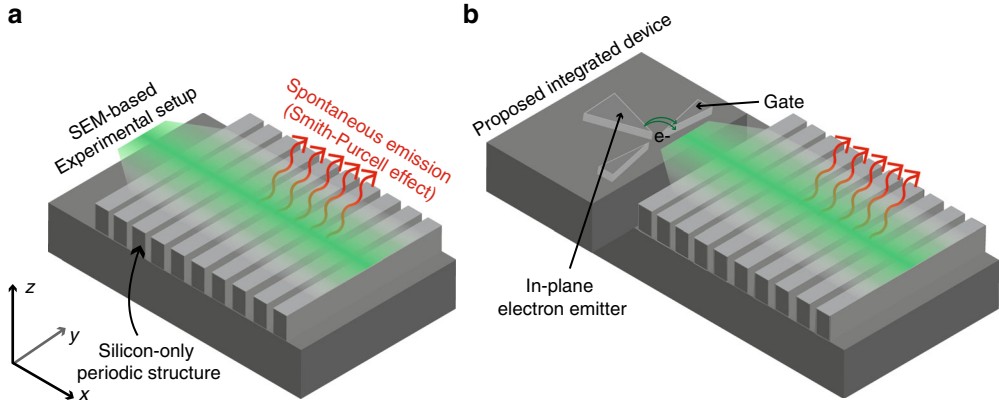

**Fig. 1** Broadly tunable radiation from all-silicon nanogratings. **a** An electron emitter (in vacuum) generates a beam of electrons traveling at a grazing angle to an all-silicon nanograting, thus generating tunable radiation that follows the SP wavelength-angle equation. In our current experimental setup (see Fig. 3), the electron emitter consists of the electron gun of a scanning electron microscope (SEM). **b** In future devices, the emitter could be integrated onto a silicon chip (see Fig. 4) with, for instance, (gated) silicon field emitter arrays. We discuss the compatibility of this proposed device with conventional fabrication techniques and CMOS-compatibility in the Discussion section and in the Supplementary Note 9

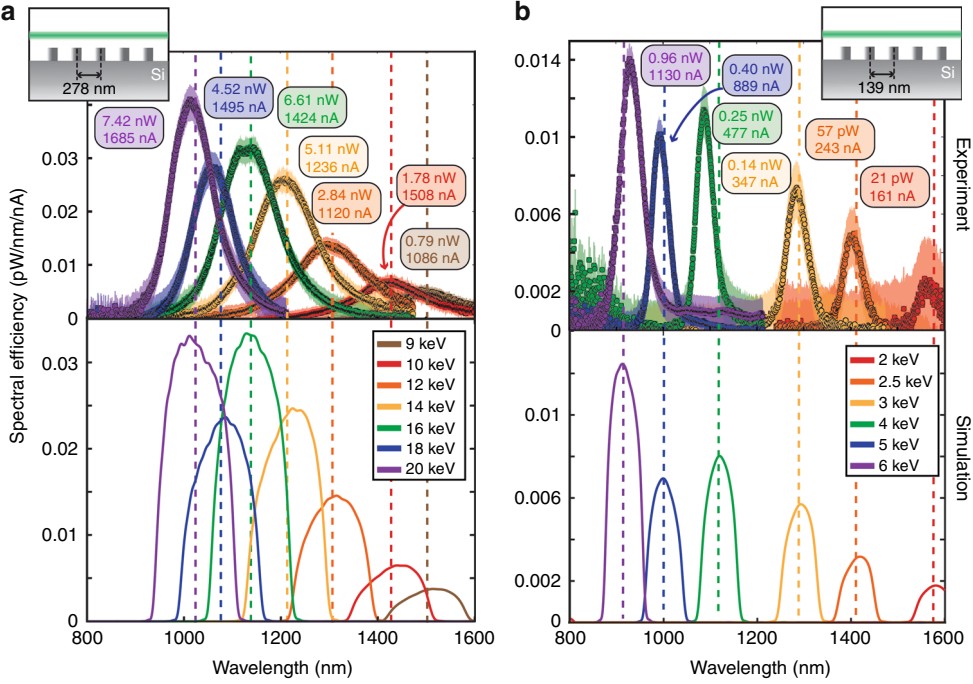

**Fig. 2** Observation of tunable radiation from silicon in the 800–1600 nm window. Experimental (top) and simulated (bottom) spectral efficiencies recorded for various kinetic energies (**a**) (9–20 keV) from 278 nm-period gratings and (**b**) (2–6 keV) from 139 nm-period gratings. The incident electron current and measured output powers are reported in the colored boxes for each measurement. Dashed lines correspond to the predicted output wavelength at normal emission direction from the SP energy-angle equation. Optical and quantum efficiencies are reported in the SI, section I. Error bars (shaded area) are estimated from the standard deviation of the background signal

lightly doped silicon substrates, which makes the sample conductive without significantly altering its optical properties. The doping thus circumvents Coulomb repulsion from charges gathering on the surface of the sample—as would happen with electrically insulating materials. The sample is bare silicon without any coating materials. Additional information on the beam (diameter and angular divergence) and sample (doping) characterization can be found in the Methods Section and in the Supplementary Note 10.

Our setup resolves the spatial, spectral and polarization behavior of emitted cathodoluminescence; spontaneous emission from free electrons is coupled out of the SEM vacuum chamber

and then goes through two arms of free space optics (1) a linear polarizer and a set of lenses focusing radiation to an optical fiber coupled to a near-infrared spectrometer and (2) a CCD camera allowing the imaging of the surface of the nanograting (of which a scanning electron micrograph is shown in Fig. 3c). After verifying that the SP signal is polarized along the direction of the electron beam (as was originally observed[26] and confirmed by Fig. 3b), we subtract from the SP signal the radiation measured at the orthogonal polarization (from radiative processes triggered by the electrons impinging on the bulk, such as electron-hole recombination and transition radiation). This definition of the background relies on the assumption that the SP electric field and

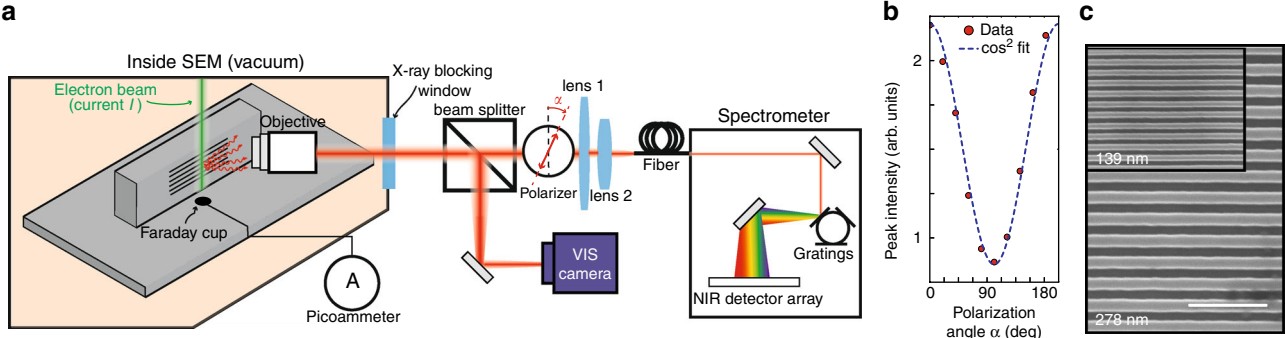

**Fig. 3** Spectrally, spatially, and polarization-resolved modified-SEM cathodoluminescence measurement setup. **a** A modified SEM generates an electron beam impinging on a nanograting at a grazing angle. A Faraday cup, mounted on the sample holder, is used to record the electron beam current $I$. A microscope objective collects and couples out the radiation induced by free electrons to a set of free space optics. **b** Peak intensity as a function of the polarization angle. $\alpha = 0$ corresponds to the direction of electron beam propagation. The dashed line shows an affine fit in $\cos^2 \alpha$. **c** Scanning-electron micrographs of 278 and 139 nm (inset) silicon nanogratings. Scale bar is 1 μm

cathodoluminescence from the bulk are incoherent—as is usually observed for low kinetic energy electrons in semiconductors[46]—and the measured background (weak incoherent cathodoluminescence in silicon[30]) is polarization-independent. We estimate the influence of this last assumption on the spectral lineshapes and power estimates, and discuss the possibility of radiation from electrons penetrating the bulk in the Supplementary Note 3. In particular, we observe some remaining incoherent cathodoluminescence around 1100 nm (corresponding to radiative recombination from silicon's indirect bandgap at 1.1 eV) that still exhibits a preferential polarization along the electron beam propagation direction (see Supplementary Fig. 4).

Our experimental setup also allows us to measure the optical power and quantum efficiencies of this free-electron-induced source. In the following, we define the power efficiency as $\eta_{\mathrm{opt}} = \frac{P_{\mathrm{opt}}}{(I\,E_{\mathrm{in}}/e)}$ where $P_{\mathrm{opt}}$ is the output optical power, $E_{\mathrm{in}}$ the incident electrons' kinetic energy and $e$ is the elementary charge. The power efficiencies of the experiments presented in Fig. 2 is in the range $6 \times 10^{-8}\,(2\,\mathrm{keV}) - 3 \times 10^{-7}\,(20\,\mathrm{keV})$. In the Supplementary Note 1, we also define the quantum efficiency of our device as the ratio of output photons to input electrons $\eta_{\mathrm{QE}} = \frac{e\lambda_0}{hc}\frac{P_{\mathrm{opt}}}{I}$ and compare our experimental results for quantum efficiency $0.02\%\,(2\,\mathrm{keV}) - 1.24\%\,(20\,\mathrm{keV})$, to other silicon-based technologies. These low power efficiencies reported in our proof-of-concept experiment naturally raise the question of the maximal power efficiency that could be achieved in a scalable on-chip device with this spontaneous emission process, which is addressed in the following section.

**Maximum power efficiency of an all-silicon free-electron source.** A fundamental metric of any system performing power conversion is its maximal power conversion efficiency. Some of the foundational principles of modern physics were used to derive such limits in the cases of, for example, heat engines (Carnot efficiency[47], based on the second principle of thermodynamics), solar cells (the Shockley-Queisser limit[48], based on fundamental considerations on recombination processes in p–n junctions). We have recently developed a framework[34] based on optical passivity[49] that allows us to compute the maximal spontaneous radiation from free-electrons interacting with an arbitrary nanophotonic medium. We apply this formalism to the system shown in Fig. 4b consisting of an electron emitter mounted onto an all-silicon nanograting, which yields the following maximal

output power

$$P \leq P_{\mathrm{max}} = \int_{\lambda_{\mathrm{min}}}^{\lambda_{\mathrm{max}}} d\lambda \frac{2\pi c}{\lambda^2} \int_0^{L_G} dx \int dS \frac{d\Gamma}{dx} \frac{hc}{\lambda} \frac{I}{S} \qquad (1)$$

with $[\lambda_{\mathrm{min}}; \lambda_{\mathrm{max}}]$ the SP bandwidth, $L_G$ the grating total length, $I$ the electron beam current, $S = \frac{\pi D^2}{4}$ its area and $\frac{d\Gamma}{dx} \propto \mathrm{MF}(\lambda)$ is the upper limit to electron spontaneous rate of emission proportional to the finite-bandwidth material factor[35] (see the full expression in the Supplementary Note 7). The material factor is defined as[35,50,51] $\mathrm{MF}(\lambda) = \frac{\omega_0}{\Delta\omega}\frac{(\epsilon-1)^2}{\epsilon} = \pi c \beta \left(\frac{1}{\beta^2} - 1\right)\frac{(\epsilon-1)^2}{\epsilon} \approx \frac{\pi c}{\beta}\frac{(\epsilon-1)^2}{\epsilon}$ for lossless materials and non-relativistic electrons ($\beta \ll 1$).

In contrast to the prevailing belief that metallic structures should yield better SP radiation efficiencies[26,32,52,53], lossless dielectrics such as silicon in the near-infrared could benefit from potentially very large quality factors from nanophotonic structure designs[54] and thus become ideal for free-electron light emitting devices, in particular for narrow-bandwidth applications. This is especially surprising since the generated optical frequencies are well above silicon's plasma frequency. At such optical frequencies, the spatial distribution of electrons can be deformed thanks to its large dielectric susceptibility. This is in strong contrast to the case of metals—the main materials used to record Smith-Purcell radiation since its original discovery[26]—where free charges screen the field of swift electrons in vacuum with a phase delay. Additional experiments on aluminum-coated silicon nanogratings—another broadly CMOS-compatible approach—are discussed in the Supplementary Note 5 and compared to our measurements on all-silicon samples, showing additional evidence of the superior material factor of lossless dielectrics for low-kinetic-energy electrons and narrow-bandwidth applications.

Our metric, based on the fundamental constraint of optical passivity, allows us to describe the maximal power conversion efficiency of an integrated SP source and provides physical insights on how to attain this upper efficiency[34]. We compare different gated FEA designs from the literature[36,37,39–41], in which intensity-voltage relations have been measured. FEA are usually characterized by their Fowler-Nordheim (FN)[38] coefficients $a_{\mathrm{FN}}$ and $b_{\mathrm{FN}}$

$$\ln \frac{I_A}{V_{GE}^2} = \ln a_{FN} - \frac{b_{FN}}{V_{GE}} \qquad (2)$$

where $I_A$ is the anode current and $V_{\mathrm{GE}}$ the gating-emitter voltage. Values of the extracted FN coefficients for the different designs are reported in the Supplementary Note 6. These five gated FEA

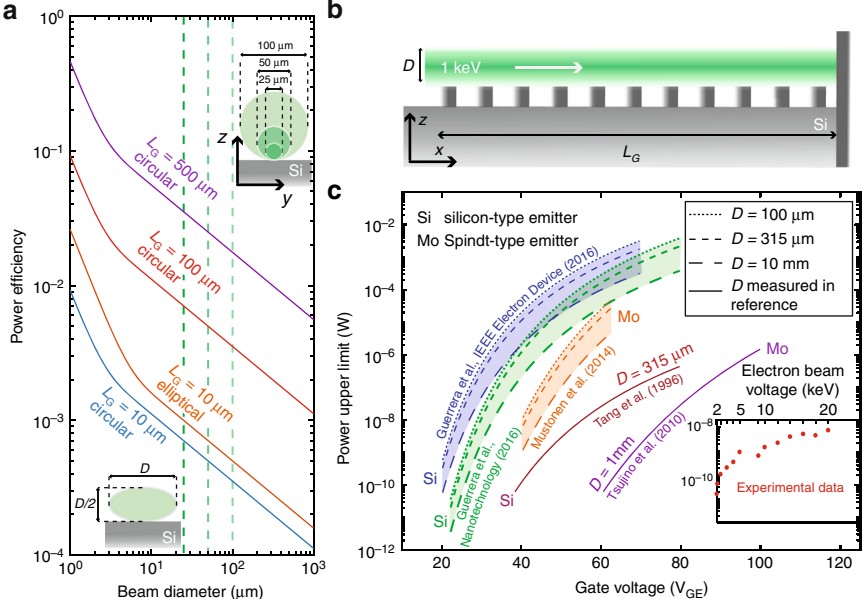

**Fig. 4** Maximal efficiency of all-silicon tunable sources in the near-infrared. **a** Calculated efficiency of a single silicon emitter integrated with a silicon nanograting as a function of the electron beam diameter. The insets show the beam shape and alignment configuration with respect to the nanograting in this simulation. **b** (Illustration) Setup used to theoretically evaluate the maximal efficiency of multiple FEA designs integrated into a SP source. **c** Calculated upper limit of the optical power as a function of the gate voltage for different FEA designs integrated with a 10 μm long silicon grating. These different designs are all gated with the same range of voltages $V_{GE}$ (20–100 V). The inset shows our experimental results in a modified SEM setup as a function of the electron beam voltage

designs are characterized at a fixed anode voltage of 1 keV, positioned at a distance $\gtrsim$ 1 mm. We assume that the electron propagates at a constant speed above the nanograting, positioned in the vicinity of the anode, which is realized if its total length $L_G$ is much smaller than the millimeter scale.

Figure 4 present the calculated maximal power efficiency from a single silicon emitter[36]. The fabrication and alignment of a gated FEA would benefit from the resolution of nanofabrication techniques: assuming the electron collimated beam geometry shown in the upper inset of Fig. 4a, one could potentially achieve maximal power efficiencies >40% by further reducing the electron beam diameter and maintaining the collimation of electron beam for long distances. However, FEAs designed to achieve smaller beam diameters usually require an additional focus voltage[40,41] which results in leakage current in the focusing gate, thus reducing the anode current.

## Discussion

To illustrate the compromise between electron beam diameter and anode current, we compare FEA designs for which the electron beam diameter is measured, with state-of-the-art high-throughput silicon and Spindt-type emitters (for the latter, we estimate the maximal power efficiency for several beam diameters). Results for the five FEA designs are reported in Fig. 4c, where the gate voltage—anode current relation is given by Eq. (2) and the radiation is centered at $\lambda = 1550$ nm. Even for very large beam diameters (10 mm), the three designs without focusing gates outperform their focused counterparts, for which the electron beam diameter has been experimentally measured (315 μm for the Spindt-type emitter[41] and 500 μm for the silicon field emitter[40]).

Silicon FEAs with state-of-the-art integration densities, longevity and high current density have recently been reported:[36,37] thanks to their large current output, they outperform their metallic counterparts[39] with similar integration densities. The ability to leverage VLSI fabrication processes for silicon FEAs is

enabling their greater scaling; here, this technological advantage of silicon FEAs finds a direct application in free-electron sources. This perspective is particularly promising for the realization of an all-silicon free-electron-driven radiation source. The proposed device in Fig. 1, made of an (in-plane) FEA integrated with a silicon nanograting on a monolithic circuit, requires some additional packaging in order to be operational. However, the development of vacuum packaging techniques[55], in addition to high-voltage DC-DC converters[56], should ensure scalability of our approach, in addition to its compatibility with CMOS fabrication processes.

In our study, the predicted maximal power efficiencies are comparable with those reported in all-silicon LED[7]. We also experimentally achieve quantum efficiencies ~1%, already comparable to all-silicon LEDs, thanks to the large energy mismatch between the incident electrons and generated photons. However, the measured power efficiencies are still several orders-of-magnitude below standard silicon sources. Achieving large power efficiencies will require phase-matching the electron excitation with a high-$Q$ resonance of the photonic structure and a small-diameter beam ($D \sim 10$ μm) with negligible angular divergence and uniform current distribution interacting with the grating over a long distance ($L_g \sim 500$ μm). Nonetheless, unlike solid-state devices, our proposal of an all-silicon integrated radiation source can leverage the advantages specific to free-electron devices to increase its power conversion efficiency, such as (1) the possibility of partially recovering the energy of electrons having interacted with the structure with a depressed collector—as with a field emitter integrated in traveling-wave tubes[57], (2) the engineering of the electron beam spatial distribution in order to increase the coupling between the beam and the nanograting (see Fig. 4a, lower inset). This engineering is enabled by the precise alignment of the electron beam with the nanograting on chip, allowed by the nanoscale resolution of CMOS fabrication techniques; or (3) pre-bunching the electron beam to facilitate stimulated emission[58,59]. The latter could be achieved with an

ultrafast FEA, such as out-of-plane silicon[60] and in-plane plasmonic[61,62] ultrafast FEAs that have recently been demonstrated. These advantages, as well as the scheme of coupling to high-Q photonic modes, makes silicon a particularly interesting platform for free-electron light sources. As silicon exhibits similar dielectric properties in the mid-infrared, we expect similar maximal power efficiencies for that wavelength range also, thus enabling applications in mid-infrared sensing; this wavelength range could also benefit from narrower bandwidths, with lower-energy electron sources[63].

The possibility of achieving stimulated emission on chip is particularly exciting, as few experiments have observed coherent emission from electrons interacting with passive systems[64,65]. Achieving stimulated emission on chip with an all-silicon structure would solve probably one of the most resilient problems of modern engineering and physics: the demonstration of an electrically pumped all-silicon laser. Another interesting aspect of free-electron sources is the possibility of engineering the structure profile in order to achieve spectral[34,66] and/or spatial[66] shaping of the emitted radiation. Adding this feature to a SP source integrated with an FEA would result in an ultra-compact device comprising (1) an on-chip free-electron source pumping (2) a nanograting, acting as an SP source whose generated radiation is (3) spatially spectrally shaped (for instance, potentially replacing bulky focusing optics at the output of the light source).

We have experimentally demonstrated a proof-of-concept, tunable, free-electron-driven radiation source made of simple silicon nanogratings and emitting optical radiation between 800–1600 nm. We have also provided a theoretical analysis of a scalable, all-silicon electron-beam light source, consisting of a silicon field emitter integrated with a silicon nanograting. In this framework we have computed the maximal power efficiency and output power that can be achieved by phase-matching the electron beam velocity with a high-Q photonic mode of the periodic structure[34]. We have also proposed several directions to further enhance the power efficiency by taking advantage of the features of free-electron sources, building upon early demonstrations of free-electron driven radiation on-chip with silicon[67] and molybdenum (Spindt-type) field emitters[68,69]. Our theoretical framework[34] is readily transferrable to other wavelength ranges and materials for which free-electron sources could also be technologically interesting, for instance in the far-infrared and THz regimes[64].

Our work suggests that free-electron-driven light-matter interactions may offer a viable way to generate light in all-silicon structures. Techniques in both electron beam physics and nanophotonic design could be leveraged to make the interaction of the electrons with the structure more efficient.

## Methods

**Experimental setup**. Our experimental setup is based on a modified Scanning Electron Microscope (SEM) and is comprised of a set of adequate free space optical components to spectrally, spatially and polarization-resolve the emission of photons from free-electrons interacting with a nanophotonic structure (see Fig. 3). We mounted a periodic sample close to parallel to the electron beam direction inside the vacuum chamber of the SEM in order to send electrons at a grazing angle with respect to the grating plane. The SEM model used for the experiment is JEOL JSM-6010LA. The SEM was operated in spot mode, in which we control precisely to position the beam so that it passes parallel to the surface near the desired area of the sample.

A Nikon TU Plan Fluor ×10 objective with a numerical aperture (NA) of 0.30 was used to collect light from the area of interest. The spectrometer used was an Acton SP-2360–2300i with a linear InGaAs photodiode detector array with detection range 0.8–1.7 μm. Monochrome images of the radiation were collected with a Hamamatsu CCD, in order to align the optical setup and spatially resolve the observed radiation. The beam currents were measured using Keithley 6485 picoammeter connected to a Faraday Cup mounted on a SEM sample holder.

**Simulation setup**. We performed time-domain simulations in order to estimate the power spectrum of photons emitted by electrons propagating at a constant

height $h$ above an all-silicon periodic structure. We designed this simulation setup in order to confirm our experimental results. We then fit the simulated power spectrum to the measured one with a single fitting parameter.

The electron beam is represented by a discrete set of dipoles with polarization along the electron direction. Each dipole is turned on at the time when the electron is flying by its position. We computed the scattered spectrum from this array of dipoles via a near-to-far-field transformation. We verified that the far-field spectrum converges when the number of dipoles was large enough.

In order to match our experimental setups with a single simulation of an electron flying at a constant height $h$ above an all-silicon grating, we made the following assumptions: (1) The corresponding height $h$ is the electron nearfield decay length $h = \frac{\gamma\beta\lambda}{4\pi}$. (2) The simulation is run for a large number of unit cells, so that the output power per unit cell has converged. (3) We can then fit our experimental results by using the total number of unit cells $N_{UC}$ as a fitting parameter. We deduced the value of the fitting angle of electron beam incidence $\theta_{fit} = 2\text{atan}\frac{h_{eff}}{LN_{UC}}$ and compared it to the experimental value in the Supplementary Note 4.

**Experimental power calibration and estimates**. In our experimental setup, we are able to measure the absolute value of the Smith-Purcell radiation by performing a calibration measurement using an AVALIGHT-HAL-CAL Calibrated VIS-Halogen Lightsource. In a reference experiment, we measured the wavelength-dependent loss function $L(\lambda)$ through our optical setup (this function encompasses absorption, reflection, the quantum efficiency of the detector, etc.). This was achieved by measuring the signal of the AVA calibrated source through our setup used to record Smith-Purcell radiation. The absolute value of the Smith-Purcell signal was obtained after normalization by $L(\lambda)$, and the detector area. Further details are given in the Supplementary Note 3.

**Sample characterization**. According to the vendor company (LightSmyth), the nanogratings were made of doped single-crystal silicon with (1, 0, 0) crystal orientation. In order to characterize the silicon nanogratings, we performed a four-point probe measurement of their resistivity on a Signatone Pro-4 Four Point Resistivity Systems. The measured resistance between the two probes is linked to the resistivity via the following equation

$$\rho = \frac{Rt}{g_4}$$

where $t = 0.7$ mm is the sample thickness and $g_4 = \frac{1}{\pi}\log\frac{\sinh(\frac{t}{s})}{\sinh(\frac{t}{2s})}$ with $s = 1.016$ mm being the distance between the two central probes.

The measured resistance for the uncoated silicon samples was $(7.42 \pm 0.19)$ Ω which corresponds to a resistivity of $(2.06 \pm 0.05)$ Ω × cm. According to doping tables, this corresponds to a doping level of $(6.74 \pm 0.2) \times 10^{15}$ cm$^{-3}$ for $p$-type silicon and $(2.24 \pm 0.06) \times 10^{15}$ cm$^{-3}$ for $n$-type silicon. In both cases, it is highly non-degenerate and the corrections to the optical properties of silicon are negligible.

**Maximum power estimates**. In Fig. 4a–c, maximum power estimates are shown for a device consisting of a field emitter array (FEA) integrated with a one-dimensional silicon grating. These estimates were obtained with Eq. (1), with the current given by the Fowler-Nordheim equation (Eq. (2) in the main text). Parameters used to obtain Fig. 4a were: beam kinetic energy of 1 keV, varying beam diameter ($x$-axis) and grating lengths (indicated on the solid lines). Parameters used to obtain Fig. 4c were: beam kinetic energy of 1 keV, grating length $L_G =$ 10 μm. The Fowler–Nordheim parameters were extracted from the references indicated. The beam diameter was varied (100 μm, 315 μm, 10 mm) when no beam measurement data was presented in the indicated reference, or was taken directly from the indicated reference when reported.

More details on the methods and any associated references are available in the Supplementary Information.

## Data availability

The data that support the plots within this paper and other findings of this study are available from the corresponding author upon reasonable request.

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

## Acknowledgements

The authors would like to acknowledge Alex Patterson and Winston Chern (MIT) for their fruitful feedback on silicon FEAs; Richard Hobbs, Chung-Soo Kim, and William Putnam, for helpful discussion regarding using dielectrics for SP radiation; Jamison Sloan, for helpful comments on the paper. Yi.Y. was partly supported by the MRSEC Program of the National

Science Foundation under Grant No. DMR-1419807. I.K. is an Azrieli Faculty Fellow, partially supported by the Azrieli Foundation and by the Seventh Framework Programme of the European Research Council (FP7-Marie Curie IOF) under grant agreement No. 328853CMC-BSiCS. K.K.B., Yu.Y., and P.D.K. gratefully acknowledge support of this project from Arthur Chu and Jariya Wanapun. P.D.K. and Yu.Y. would like to acknowledge support by the Air Force Office of Scientific Research (AFOSR) grant under contract NO. FA9550-19-1-0065. This material is based upon work supported in part by the U.S. Army Research Laboratory and the U.S. Army Research Office through the Institute for Soldier Nanotechnologies, under contract number W911NF-18–2–0048.

## Author contributions

C.R.-C., S.K., A.M., I.K., K.K.B., and M.S. conceived the project. C.R.-C. and Y.Y. developed the numerical methods. C.R.-C., S.K., Yi.Y., and A.Z. performed the experiment. Yu.Y. prepared the aluminum-coated samples. C.R.-C., S.K., Yi.Y. and P.D.K. developed the theoretical model and discussion around FEA integrated with SP sources. J.D.J., K.K.B., I.K., and M.S. supervised the project. C.R.-C. and S.K. wrote the paper with inputs from all authors.

## Additional information

**Competing interests:** The authors declare the following patent: U.S. Patent Application No.: US20180287329A1 (Ref[70]).

