## [Peer Review File · Nature Communications]

Reviewers' Comments:

Reviewer #1:

Remarks to the Author:

The manuscript reports on the experimental demonstration of near-infrared Smith-Purcell (SP) emission from silicon nano-gratings, and on an associated analytical/computational study of integrated 'all-silicon' SP light sources. Both the experimental and theoretical components of this work have evidently been very carefully and rigorously executed, and have generated a comprehensive set of novel and interesting results. However, the manuscript as currently constructed does not do justice to this work, mainly because far too much content has been relegated to the Supplementary Information (SI) file:

The SI file should contain only additional information that may be of interest to readers but which is not essential to understanding of the main manuscript. In other words, the main manuscript should be a fully self-contained and understandable document without the SI file – it is not. In particular the manuscript must contain a methods section covering details of sample fabrication, experimental and computational/analytical methods, and the various assumptions and parameter values that underpin Fig. 4.

With appropriate restructuring of the existing content between the manuscript and SI files I would be pleased to recommend publication in Nature Communications.

On some minor points:

- (Abstract) The V in VLSI stands for 'very' not 'ultra'.
- (Page 3) While it is certainly true that prior studies of SP emission have almost exclusively considered metallic grating structures, it is not true to say that that dielectric and semiconductor substrates have been ignored in regard more generally to "electron-driven radiation", i.e. including light emission induced by electron impact (e.g. J. Appl. Phys. 115, 244307 (2014) [Ref. 44]; Appl. Phys. Lett. 113, 241902 (2018); ACS Photonics 5, 1381 (2018); Phys. Rev. B 97, 081404(R) (2018)). This paragraph should carefully distinguish between the general case and the more specific case of grazing incidence (SP-type) interactions.
- (Fig. 2) The simulations have been designed to mimic the experimental situation as closely as possible, but I wonder whether reasons for the different lineshapes between experimental and simulated spectra are understood? Also, in regard to this figure, the coloured boxes containing nW/nA values are not explained or referred to at any point – would they form part of an explanation as to why the peaks centred at ~1080 nm for the 278 nm grating and at 1000 nm for the 139 nm grating break from the general trend of increasing peak height with acceleration voltage?
- (Page 8) Detail on experimental setup and calibration procedures currently in SI largely resolve the problem, but the statement beginning "we subtract from the SP signal the radiation measured at the orthogonal polarization" is somewhat open to misinterpretation: I initially took this to be saying that background emission polarized parallel to SP emission was being ignored rather than that a background level measured for the orthogonal polarization was being assumed to exist across all polarizations.

Reviewer #2:

Remarks to the Author:

This paper presents new experimental results demonstrating generation of Smith Purcell radiation in the infrared using sub-wavelength silicon gratings. The other major component of the paper is a theoretical parameter study on maximal radiation generation and efficiency capabilities for this

technique, with the aim in mind of making fully on-chip radiation sources using field emitter arrays to generate the electrons. A subset of these authors have previously presented Smith Purcell radiation results using metallic gratings in what appears to be the same or similar apparatus. Smith Purcell radiation is hardly a new discovery and swapping out a metallic grating for a silicon one may seem like a trivial change to produce another paper on the subject. However, exploration of SP from dielectric and semiconductor materials is an interesting topic for the potential production of integrated on-chip radiation sources. The paper's value is enhanced by the fact that the authors have performed a very careful and well presented analysis that includes calibrated absolute measurement of the emitted power spectrum of the gratings, reproduced these through simulation, and have put these results in a useful context via the subsequent theoretical analysis. They also notably compare their results in the supplemental section with the same gratings coated with Aluminum, showing very similar power emission, which further supports the potential for silicon as a useful radiator. I find the paper on the whole to be well written and highly interesting. Whether the novelty of the work rises to the standards of Nature Communications I will leave to the editors to decide. But I think it is a valuable contribution to the scientific literature and should certainly be published. The theoretical parameter study component of the work makes use of the theoretical formulation from one of their prior papers [Ref 30] to determine maximal radiation power and to estimate power efficiency (relative to the incident electron beam power). I have only a few additional minor comments:

1. The beam current " I " should be defined in proximity to its first appearance adjacent to Figure 3.
2. The authors note the narrower bandwidths for case (b) compared with case (a) in Figure 2, but don't explain why this is so. Is this simply a function of the angular aperture of the collecting optics combined with the SP energy-angle relation or is there a more sophisticated explanation?

Response Letter to the Reviewers

We are grateful for the constructive comments from all the reviewers on our manuscript (NCOMMS-19-09341 – entitled “**Towards integrated tunable all-silicon free-electron light sources**”).

In the text below each of the comments from each reviewer is quoted in *Italic* and is followed by the corresponding detailed response. We also revised the manuscript and the supplementary material accordingly. All modifications are shown in **blue** in these documents.

General comments from Reviewer #1

Reviewer #1 – Comment #1

The manuscript reports on the experimental demonstration of near-infrared Smith-Purcell (SP) emission from silicon nano-gratings, and on an associated analytical/computational study of integrated ‘all-silicon’ SP light sources. Both the experimental and theoretical components of this work have evidently been very carefully and rigorously executed, and have generated a comprehensive set of novel and interesting results.

Response from the Authors: We would like to thank Reviewer #1 for his/her appreciation of the novelty and rigor our work.

Reviewer #1 – Comment #2

However, the manuscript as currently constructed does not do justice to this work, mainly because far too much content has been relegated to the Supplementary Information (SI) file:

The SI file should contain only additional information that may be of interest to readers but which is not essential to understanding of the main manuscript. In other words, the main manuscript should be a fully self-contained and understandable document without the SI file – it is not. In particular the manuscript must contain a methods section covering details of sample fabrication, experimental and computational/analytical methods, and the various assumptions and parameter values that underpin Fig. 4.

Response from the Authors: We thank Reviewer #1 for his/her suggestions on the form of this manuscript, that we have taken into account to write this revised version. We have added a Methods section at the end of the revised manuscript which includes:

- Details on the experimental setup
- Details on the simulation setup
- Method for experimental power calibration
- Sample characterization
- Method and main assumptions for maximum power estimates

With these elements added to the revised main text, we believe we now have a self-contained manuscript where the reader can find all relevant details to the understanding of our work. Details that are not essential to understand the main text are kept in the Supplementary Information, such as our additional measurements on aluminum-coated gratings and beam characterization.

Reviewer #1 – Comment #3

With appropriate restructuring of the existing content between the manuscript and SI files I would be pleased to recommend publication in Nature Communications.

Response from the Authors: We thank Reviewer #1 for his/her positive feedback on the work and hoped that we have addressed his comment on the form of the manuscript.

Specific comments from Reviewer #1

Reviewer #1 – Comment #4

On some minor points:

- (Abstract) The *V* in *VLSI* stands for ‘very’ not ‘ultra’.

Response from the Authors: We thank Reviewer #1 for spotting this typo and have carefully checked the use of this acronym throughout the main text and Supplementary Information.

Reviewer #1 – Comment #5

- (Page 3) *While it is certainly true that prior studies of SP emission have almost exclusively considered metallic grating structures, it is not true to say that that dielectric and semiconductor substrates have been ignored in regard more generally to “electron-driven radiation”, i.e. including light emission induced by electron impact (e.g. J. Appl. Phys. 115, 244307 (2014) [Ref. 44]; Appl. Phys. Lett. 113, 241902 (2018); ACS Photonics 5, 1381 (2018); Phys. Rev. B 97, 081404(R) (2018)). This paragraph should carefully distinguish between the general case and the more specific case of grazing incidence (SP-type) interactions.*

Response from the Authors: We would like to thank Reviewer #1 for bringing this point to our attention. We have carefully rewritten this paragraph and added all the suggested references:

“Free-electron sources^{1–3}, especially in the context of Smith-Purcell (SP) radiation⁴, are natural candidates to address this challenge, thanks to their exceptional tunability. However, Smith-Purcell radiation from dielectric substrates, let alone silicon, has not been utilized so far. In contrast, a related effect, the laser acceleration of particles interacting with confined modes in dielectric structures⁵, has been widely studied. More recently, there has been a growing interest in the study of incoherent⁶ and coherent⁷ (transition radiation) cathodoluminescence in dielectrics and semiconductors, with novel experimental techniques to disentangle their relative importance^{8,9}. This lack of work in dielectrics and semiconductors for radiation generation from free electrons incident at a grazing angle¹⁰ may originate from the fact that the SP effect was first observed in metallic gratings and was subsequently explained as the constructive interference from the periodic motion of free currents along the surface¹¹, or as the motion of image charges in a perfect conductor⁴.”

Reviewer #1 – Comment #6

- (Fig. 2) *The simulations have been designed to mimic the experimental situation as closely as possible, but I wonder whether reasons for the different lineshapes between experimental and simulated spectra are understood? Also, in regard to this figure, the coloured boxes containing nW/nA values are not explained or referred to at any point – would they form part of an explanation as to why the peaks centred at ~1080 nm for the 278 nm grating and at 1000 nm for the 139 nm grating break from the general trend of increasing peak height with acceleration voltage?*

Response from the Authors: Even though we have designed our numerical simulations to mimic the experimental setup, there remain some differences that may explain the different lineshapes. In particular, the angular near-to-farfield transformation used in our numerical

simulations assumes a Heaviside window function $[-\theta_{\max}; \theta_{\max}]$, where $\theta_{\max} = 17.5$ deg is the half-collection angle of the objective used in our experiment. This may explain the relatively sharp edges of the lineshapes from our numerical simulations, since the spectral lineshape and angular distributions are directly related with the Smith-Purcell dispersion relation.

However, typical angular transmission will be a smoother function of the incident angle, thus resulting in a smoothing of the spectral lineshape. For reference, below we plot the Smith-Purcell signal from a 20 keV electron beam impinging on a 278nm grating, modulated by a Heaviside window angular transmission and by a Gaussian window angular transmission. We speculate that the actual transmission of our objective lies between these two cases.

Figure 1: Comparison of the influence of various angular transmission profiles. The full spectrum is assumed to be a Lorentzian peaked at $\lambda_0 = L/\beta$ and width $\Delta\lambda = L$.

Regarding the second point raised by Reviewer #1: we could observe remaining incoherent cathodoluminescence signal around 1100 nm (1.1 eV), even after polarization-sensitive background subtraction. This hints at a partial polarization preference from the incoherent cathodoluminescence background. We can see from the data in Figure S4 from the Supplementary Information that this results in a $\sim 10\%$ overestimation of the signal when it is spectrally superposed with the incoherent cathodoluminescence background. This explains the fact that the measurement at 16 keV (278 nm period grating) and at 4 keV (139 nm period grating) break the general trend of decreasing peak height with decreasing acceleration voltage. This speculative explanation is also supported by the fact that the measurements with aluminum-coated gratings (see Figure S7 in the Supplementary Information) show a slightly smoother tendency (we still observe an incoherent cathodoluminescence background around 1100 nm, but it is mitigated by the superficial layer of aluminum which also participates in the electron nearfield scattering).

Reviewer #1 – Comment #7

- (Page 8) Detail on experimental setup and calibration procedures currently in SI largely resolve the problem, but the statement beginning “we subtract from the SP signal the radiation measured at the orthogonal polarization” is somewhat open to misinterpretation: I initially took this to be saying that background emission polarized parallel to SP emission

was being ignored rather than that a background level measured for the orthogonal polarization was being assumed to exist across all polarizations.

Response from the Authors: We thank Reviewer #1 for bringing up this ambiguity and have carefully rewritten the paragraph:

“This definition of the background relies on the assumption that the SP electric field and cathodoluminescence from the bulk are incoherent – as is usually observed for low-kinetic energy electrons in semiconductors – and the measured background (weak incoherent cathodoluminescence in silicon) is polarization-independent. We estimate the influence of this last assumption on the spectral lineshapes and power estimates, and discuss the possibility of radiation from electrons penetrating the bulk in Section III of the SI. In particular, we observe some remaining incoherent cathodoluminescence around 1,100 nm (corresponding to radiative recombination from silicon’s indirect bandgap at 1.1 eV) that still exhibits a preferential polarization along the electron beam propagation direction (see Figure S4).”

General comments from Reviewer #2

Reviewer #2 – Comment #1

This paper presents new experimental results demonstrating generation of Smith Purcell radiation in the infrared using sub-wavelength silicon gratings. The other major component of the paper is a theoretical parameter study on maximal radiation generation and efficiency capabilities for this technique, with the aim in mind of making fully on-chip radiation sources using field emitter arrays to generate the electrons. A subset of these authors have previously presented Smith Purcell radiation results using metallic gratings in what appears to be the same or similar apparatus. Smith Purcell radiation is hardly a new discovery and swapping out a metallic grating for a silicon one may seem like a trivial change to produce another paper on the subject. However, exploration of SP from dielectric and semiconductor materials is an interesting topic for the potential production of integrated on-chip radiation sources. The paper’s value is enhanced by the fact that the authors have performed a very careful and well-presented analysis that includes calibrated absolute measurement of the emitted power spectrum of the gratings, reproduced these through simulation, and have put these results in a useful context via the subsequent theoretical analysis. They also notably compare their results in the supplemental section with the same gratings coated with Aluminum, showing very similar power emission, which further supports the potential for silicon as a useful radiator. I find the paper on the whole to be well written and highly interesting. Whether the novelty of the work rises to the standards of Nature Communications I will leave to the editors to decide. But I think it is a valuable contribution to the scientific literature and should certainly be published. The theoretical parameter study component of the work makes use of the theoretical formulation from one of their prior papers [Ref 30] to determine maximal radiation power and to estimate power efficiency (relative to the incident electron beam power). I have only a few additional minor comments:

Response from the Authors: We would like to thank Reviewer #2 for his/her appreciation of the value of our work.

Specific comments from Reviewer #2

Reviewer #2 – Comment #2

1. The beam current “I” should be defined in proximity to its first appearance adjacent to Figure 3.

Response from the Authors: In the revised version of the manuscript, we have reminded the reader of the beam current I in the (1) caption of Figure 3 and (2) in Figure 1.

Reviewer #2 – Comment #3

2. The authors note the narrower bandwidths for case (b) compared with case (a) in Figure 2, but don't explain why this is so. Is this simply a function of the angular aperture of the collecting optics combined with the SP energy-angle relation or is there a more sophisticated explanation?

Response from the Authors: As suggested by Reviewer #2, the bandwidth narrowing for slow electrons can be explained directly from the Smith-Purcell dispersion relation, assuming the angular collection is large enough to observe it.

For an angular collection $[-\theta_{\max}; \theta_{\max}]$, the maximum and minimum wavelengths can be derived from the Smith-Purcell dispersion relation $\lambda_{\pm} = L(\frac{1}{\beta} \pm \sin \theta_{\max})$. The relative bandwidth scales like $\frac{\Delta\lambda}{\lambda_0} = 2\beta$. This explains the bandwidth narrowing for slow electrons. We have added a sentence explaining this observation in the revised main text.

References:

1. Friedman A, Gover A, Kurizki G, Ruschin S, Yariv A. Spontaneous and stimulated emission from quasifree electrons. *Rev Mod Phys*. 1988;60(2):471-535. doi:10.1103/RevModPhys.60.471
2. Adamo G, MacDonald KF, Fu YH, et al. Light well: A tunable free-electron light source on a chip. *Phys Rev Lett*. 2009;103(11). doi:10.1103/PhysRevLett.103.113901
3. McNeil BWJ, Thompson NR. X-ray free-electron lasers. *Nat Photonics*. 2010;4(12):814-821. doi:10.1038/nphoton.2010.239
4. Smith SJ, Purcell EM. Visible light from localized surface charges moving across a grating. *Phys Rev*. 1953;92(4):1069. doi:10.1103/PhysRev.92.1069
5. England RJ, Noble RJ, Bane K, et al. Dielectric laser accelerators. *Rev Mod Phys*. 2014;86(4):1337-1389. doi:10.1103/RevModPhys.86.1337
6. Feldman MA, Dumitrescu EF, Bridges D, et al. Colossal photon bunching in quasiparticle-mediated nanodiamond cathodoluminescence. *Phys Rev B*. 2018;97(8):081404. doi:10.1103/PhysRevB.97.081404
7. Clarke BP, Gholipour B, MacDonald KF, Zheludev NI. All-dielectric free-electron-driven holographic light sources. *Appl Phys Lett*. 2018;113(24):241902. doi:10.1063/1.5048503
8. Brenny BJM, Coenen T, Polman A. Quantifying coherent and incoherent cathodoluminescence in semiconductors and metals. *J Appl Phys*. 2014;115(24):244307. doi:10.1063/1.4885426
9. Mignuzzi S, Mota M, Coenen T, et al. Energy–Momentum Cathodoluminescence Spectroscopy of Dielectric Nanostructures. *ACS Photonics*. 2018;5(4):1381-1387. doi:10.1021/acsp Photonics.7b01404
10. Ohtaka K. Smith-Purcell radiation from metallic and dielectric photonic crystals. In: *Technical Digest. CLEO/Pacific Rim 2001. 4th Pacific Rim Conference on Lasers and Electro-Optics (Cat. No.01TH8557)*. Vol 1. IEEE; :I-272-I-273. doi:10.1109/CLEOPR.2001.967830
11. Van den Berg PM. Smith–Purcell radiation from a point charge moving parallel to a reflection grating. *J Opt Soc Am*. 1973;63(12):1588. doi:10.1364/JOSA.63.001588

Reviewers' Comments:

Reviewer #1:

Remarks to the Author:

The authors have suitably restructured the manuscript and provided a detailed response to all review comments. I am happy to recommend publication.

Reviewer #2:

Remarks to the Author:

The authors have adequately responded to the comments of my prior report. I recommend the manuscript for publication.

Response Letter to the Reviewers

We are grateful for the constructive comments from all the reviewers on our manuscript (NCOMMS-19-09341A – entitled “**Towards integrated tunable all-silicon free-electron light sources**”).

In the text below each of the comments from each reviewer is quoted in *Italic* and is followed by the corresponding detailed response.

General comments from Reviewer #1

Reviewer #1 – Comment #1

The authors have suitably restructured the manuscript and provided a detailed response to all review comments. I am happy to recommend publication.

Response from the Authors: We would like to thank Reviewer #1 for his/her appreciation of our response and for recommending publication.

General comments from Reviewer #2

Reviewer #2 – Comment #1

The authors have adequately responded to the comments of my prior report. I recommend the manuscript for publication.

Response from the Authors: We would like to thank Reviewer #2 for his/her appreciation of our response and for recommending publication.